# Analysing the impact of COVID-19 risk perceptions on route choice behaviour in train networks

**Sanmay Shelat** [1]*, **Thijs van de Wiel** [1], **Eric Molin**[2], **J. W. C. van Lint**[1], **Oded Cats**[1]

**1** Department of Transport and Planning, Delft University of Technology, Delft, Netherlands, **2** Transport and Logistics Group, Delft University of Technology, Delft, Netherlands

* s.shelat@tudelft.nl

## Abstract

**Data Availability Statement:** All stated COVID-19 risk rating and train route choice data are available from https://doi.org/10.4121/16896913.v1.

**Funding:** SS was funded by the My-TRAC (H2020 Grant No. 777640) project and the Amsterdam

### Introduction

Unlike previous pandemics, COVID-19 has sustained over a relatively longer period with cyclical infection waves and numerous variants. Public transport ridership has been hit particularly hard. To restore travellers' confidence it is critical to assess their risk determinants and trade-offs.

### Methods

To this end, we survey train travellers in the Netherlands in order to: (i) quantify the impact of trip-specific, policy-based, and pandemic-related attributes on travellers' COVID-19 risk perceptions; and (ii) evaluate the trade-off between this risk perception and other travel attributes. Adopting the hierarchical information integration approach, in a two-stage stated preference experiment, respondents are asked to first rate how risky they perceive different travel situations to be, and then to choose between different travel options that include their own perceived risk rating as an attribute. Perceived risk ratings and choices between travel options are modelled using a linear regression and a mixed multinomial logit model, respectively.

### Results

We find that on-board crowding and infection rates are the most important factors for risk perception. Amongst personal characteristics, the vulnerability of family and friends has the largest impact—nearly twice that of personal health risk. The bridging choice experiment reveals that while values of time have remained similar to pre-pandemic estimates, travellers are significantly more likely to choose routes with less COVID-19 risk (e.g., due to lower crowding). Respondents making longer trips by train value risk four times as much as their shorter trip counterparts. By combining the two models, we also report willingness to pay for mitigating factors: reduced crowding, mask mandates, and increased sanitization.

Institute for Advanced Metropolitan Solutions. The Transport Institute of TU Delft provided funding for the survey. The funders had no role in study design, data collection and analysis, decision to publish, or preparation of the manuscript.

**Competing interests:** The authors have declared that no competing interests exist.

## Conclusion

Since we evaluate the impact of a large number of variables on route choice behaviour, we can use the estimated models to predict behaviour under detailed pandemic scenarios. Moreover, in addition to highlighting the importance of COVID-19 risk perceptions in public transport route choices, the results from this study provide valuable information regarding the mitigating impacts of various policies on perceived risk.

## 1. Introduction

The coronavirus pandemic (COVID-19) has drastically disrupted public and social life around the world. Governmental and individual interventions to prevent transmission have led to unprecedented changes in travel patterns [1, 2]. While vaccination efforts have begun recently, social-distancing, improving ventilation, and masking have been the main strategies employed against the virus, which primarily spreads via respiratory droplets and aerosols [3]. Public transport, which seems to be inherently incompatible with these defences, has been hit particularly hard. In the Netherlands, a boarding reduction of nearly 90% was observed in April 2020, the first month of strict lockdown measures [4]. While travellers gradually found their way back on the trains, the demand relative to the previous year peaked at about 50% in July 2020 and then lowered towards the end of the year as cases (and consequently restrictions) peaked in December 2020 [5]. This decline in ridership is in part due to an overall reduction in travel demand as people worked from home and avoided public places. Travellers' perceptions of COVID-19 risk, often fuelled by advisories against public transport use, also contribute to the reduction in ridership. We explore this latter component in this study by explicitly quantifying the impact of different aspects of public transport travel and other contextual factors on travellers' COVID-19 risk perceptions through a two-stage stated choice experiment.

Authorities and operators must work towards ensuring that poor perceptions of public transport and a modal shift towards private automobiles do not become lasting consequences of the COVID-19 pandemic. The sustained global impact of COVID-19 and reviving fears with cyclical infection waves and new variants means that fear of the virus is not likely to disappear immediately even after vaccines are widely available. Therefore, to restore confidence in public transport, it is critical to assess the degree to which different factors contribute to the perceived risk of contracting COVID-19 on public transport. Furthermore, how large this perceived risk looms in travellers' minds and affects their travel choices is also important. However, most studies on COVID-19 and previous pandemics have only studied perceived risks in public transport through simple Likert scale questions and descriptive statistics, such as mode share or ridership drops.

The contribution of this work is twofold: (i) explicitly quantifying the impact of different aspects of public transport travel and other contextual factors on travellers' COVID-19 risk perceptions and (ii) subsequently evaluating the trade-off between this risk perception and other travel attributes. We adopt the hierarchical information integration approach [6, 7] whereby a two-stage stated choice experiment is conducted. First, respondents are asked to rate how risky they perceive different travel situations to be, and second, they are asked to choose between different travel options that include their own perceived risk rating as an attribute. We conduct a survey with train travellers in the Netherlands in December 2020, just before the imposition of a stricter lockdown. Apart from highlighting the importance of COVID-19 risk perceptions in public transport route choices, this study provides valuable information regarding the mitigating impacts of various policies on perceived risk.

In the next section, we present a review of studies on public transport risk perceptions. In the following methodology section, we give a background of the hierarchical information integration approach, detail the experiment design implemented, and explain our model estimation procedure. The results are then presented before concluding with a discussion of the results, limitations, and potential policy implications.

## 2. Literature review

Previous studies have analysed the perceived risk of travelling on public transport primarily in the context of terror attacks and pandemics. After the September 11 attacks in New York City, Holguín-Veras et al. [8] found that a significant proportion of inter-city travellers reported to be more conscious of security, other travellers, and their mode choice. Stated choices indicated that stress due to the attacks as well as how much the event affected their choice to travel had a significant impact on mode choice. Elias et al. [9] analysed the fear of terror attacks on public transport in two Israeli cities and its impact on mode choice. They found that while the fear of being involved in a terror attack was (in line with reality) lower than that of being involved in a road crash, a significant proportion of respondents reported that they had or would refrain from using buses because of such threats.

While terror attacks and threats do not always imply that another attack is imminent, with disease outbreaks the risk is persistent and often accompanied by official advice to avoid public transport. In fact, some authors have interpreted a loss in public transport ridership as a proxy for epidemic fear. Wang [10] found an immediate drop in daily ridership and a lag in how quickly ridership returns with each additional case of the 2003 SARS epidemic in Taipei City. These were interpreted as 'fresh' and 'residual' fear of the epidemic, respectively. Kim et al. [11] also used reduction in ridership to assess the fear of the MERS outbreak in Seoul. Lau et al. [12] collected SARS related perceptions and behaviours at different stages of that pandemic in Hong Kong and found that avoiding public transport was increasingly seen as an effective method of disease prevention in the initial phase. In the second phase (when the number of new cases was decreasing), public transport avoidance was seen to be less effective and became less common. The trend for risk perception of transmission in public transport was found to follow that of the daily number of new cases. In a similar study for the 2009 swine flu outbreak, Rubin et al. [13] found that about half of their respondents believed that avoiding public transport was effective in preventing transmission which also made them more likely to have actually taken such measures.

Given the unprecedented spatial and temporal scale of COVID-19, a significantly larger body of literature is available for this pandemic. The impact of COVID-19 on travel behaviour—particularly overall travel reduction and drop in public transport mode share—has been widely studied (e.g., Bucsky [14], Engle et al. [15], Teixeira and Lopes [16]). Surveying respondents from Germany, Gerhold [17] found that the vast majority did not feel safe in public transport and were avoiding it. In Australia, after restrictions were eased concerns regarding travelling in public transport lowered from the levels seen in the early months of the pandemic but continued to remain higher than normal [18]. Similar reports of high risk perception are available from other countries, including India [19], Pakistan [20], Turkey [21], and Netherlands [22]. Despite the high risk perceptions, some have noted the lack of outbreaks linked to public transport and suggest that if recommended measures are implemented the risk of contracting COVID-19 as a passenger could actually be lower than many other settings [23–25]. Empirical findings from recent research remain inconclusive. While, in an epidemiological study, Hu et al. [26] found that the risk of contracting COVID-19 in trains is high, studies investigating whether the coronavirus could be detected on frequently touched surfaces or in the air in public transport vehicles in Barcelona and London found little to no trace of the virus [27, 28].

The vast majority of studies mentioned above analysed COVID-19 risk perceptions on public transport via descriptive statistics on a few Likert scale questions, ridership drops, or aggregate mode share. While some have correlated socio-economic attributes with risk perception (e.g., Beck and Hensher [18], Kim et al. [11]), only few have assessed the impacts of different risk determinants on the overall risk perception or how travellers trade-off these factors when making choices. Cho and Park [29] compared choice estimates from survey data collected before and after the outbreak of COVID-19 in Seoul and estimated that crowding impedance was 1.04–1.23 times higher during the pandemic. Shelat et al. [30] performed a more comprehensive choice analysis to assess behaviour related to three criteria affecting transmission risk: crowding, exposure duration, and prevalent infection rate. Using latent class choice modelling they discovered two segments amongst Dutch travellers—one concerned about the pandemic and one largely indifferent towards it. The former group had a significantly higher crowding valuation, sought to sit away from others, and was more sensitive to changes in infections rates.

The impact of COVID-19 safety measures has been analysed by Awad-Núñez et al. [31] and Aaditya and Rahul [32] in Spain and India, respectively. Awad-Núñez et al. [31] explored the willingness to use and pay more for public transport if such measures were implemented. They found that increasing supply to avoid crowding and improving on-board cleaning resulted in a higher willingness to use public transport. Aaditya and Rahul [32] developed a structural model for safety perceptions using safety measures within an integrated choice and latent variable model. Although they were unable to estimate how safety perception was traded-off with other variables, they found that higher safety perception increased willingness to use public transport.

Our study adds to the scarce literature analysing of travellers' risk perceptions and choice behaviour in the age of COVID-19. Instead of directly measuring the impact of COVID-19 related variables on choice behaviour (as Shelat et al. [30] do), we conceptualize these variables as affecting travellers' perceived risk, which, in turn, affects their behaviour. This is in line with the notion that decision-makers tend to utilise perceptions of reality rather than various individual objective measures, particularly for abstract concepts such as risk and safety [33]. With the approach adopted here, we are able to measure the impact of a larger number of variables without overloading respondents with information and cognitive effort. In addition to the safety measures considered by Aaditya and Rahul [32], we also account for broader contextual variables (e.g., lockdown status) and qualitative indicators (e.g., infection anxiety). Moreover, we are able to conclude which factors affect travellers' perceived risk as well as the willingness to pay to reduce such perceived risk.

## 3. Methods and materials

In this section, we first present the hierarchical information integration approach and how it is used in this study. Next, we discuss the design of the risk perception rating sub-experiment and route choice experiment, followed by an outline of the personal characteristics collected. Finally, after presenting data collection and sample characteristics, we discuss estimation procedures for the two experiments.

### 3.1. Concept

When presented with complex decision problems in stated preference experiments, subjects may suffer from information overload and excessive cognitive burden, which can compromise the validity of their responses. The hierarchical information integration approach developed by Louviere [7] models such judgement and decision making problems involving a large number of attributes. It assumes that individuals process information in an hierarchical manner: first, attributes are grouped into logical, functional, or other subsets; these attribute subsets are

then evaluated to define the value of the decision constructs they represent; and finally, the separate decision constructs are integrated into an overall judgement or preference [7, 34]. This assumption allows the researcher to divide complex problems into smaller sub-experiments analysing different decision constructs which are then linked by an overall bridging experiment. The bridging experiment includes all the decision constructs as attributes (whose levels are defined by the sub-experiments) and evaluates their part-worth utilities in order to reach an overall judgement or choice.

We apply a variant of the conventional hierarchical information integration approach that has been used in a number of transportation studies (e.g., [33, 35]). The bridging experiment in this flavour of hierarchical information integration includes physical attributes in addition to decision constructs from sub-experiments [6]. As shown in Fig 1, our study has a single decision construct, risk perception, which is defined in a rating-based sub-experiment. Many of the variables contributing to risk perception would, logically, not vary across alternatives. Thus, in a typical stated preference experiment they would simply become contextual attributes. However, with our approach they can be encapsulated within a risk rating which can then be different across alternatives in the bridging choice experiment. As they are separate experiments, we can also expect respondents to understand this variation in risk ratings intuitively. Risk perception and two additional variables, travel time and cost, serve as attributes of train alternatives in the choice-based bridging experiment. Travel time and cost are typical parameters in transportation choice experiments and have been consistently found to have significant impacts on behaviour. Travel cost was included separately to be able to obtain monetary trade-offs with COVID-19 related variables. Travel time was included as an alternative attribute rather than a contributor to risk perception in terms of exposure duration (because Shelat et al. [30], examining a similar sample and choice situation, did not find a significant effect of the latter). Incidentally, the inclusion of travel time as a separate variable also allows us to compare value-of-time with pre-pandemic estimates.

## 3.2. Experiment

**3.2.1. Risk perception.** As noted previously, in the first part of the experiment, travellers' COVID-19 related risk perception is observed in a rating-based sub-experiment. Respondents

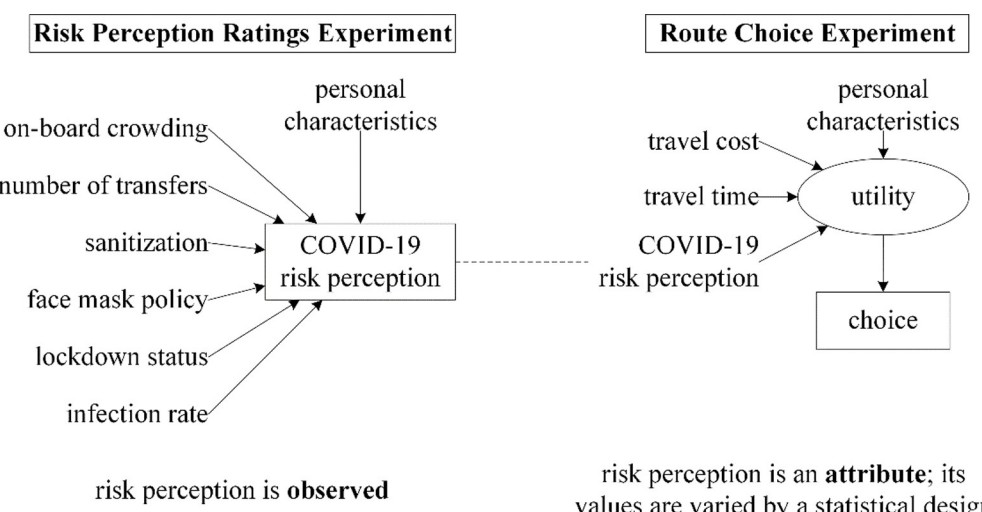

**Fig 1. Hierarchical information integration framework used in this study (illustration adopted from [33]).**

are asked to evaluate their perceived risk of getting infected with the coronavirus on a scale from 1 (low) to 5 (high) for a series of train trips characterised by attributes relevant to risk perception. We sought to include trip-specific (which could be controlled by either the traveller or operator), policy-based (which are implemented by the operator or government), and pandemic-context defining (over which direct control is not possible) attributes. From these categories, to avoid overloading respondents, we limited ourselves to a total of six attributes that we expected (partly based on literature) to have the highest impact on perceived risk.

*Trip specific attributes*. 1. On-board crowding: Reducing exposure to other people is the primary defence against COVID-19 [3] but it is largely incompatible with the idea of public transport such as trains. While on-board crowding is disliked even under normal circumstances [36], Shelat et al. [30] find this effect to be pronounced manifold during the COVID-19 pandemic, especially amongst travellers who are concerned about the virus. We describe on-board crowding conditions to respondents using seat occupancy and the possibility to sit alone, particularly because we expect the latter to be a significant contributor to risk perception. Four levels are included: almost empty (10% seat occupancy), able to sit alone (30%), not able to sit alone (60%), and standing room only (100%). We note that no seat blocking was in place in Dutch public transport systems at the time of the survey.

2. Transfers: When boarding or alighting, travellers tend to gather at train doors, increasing exposure to one another. Thus, a higher number of transfers could increase risk perception even when the crowding on-board is relatively low. Moreover, the exposure to new (unknown) people with each transfer could also increase the sense of risk. Since the proportion of trips reduces quickly with each transfer we include only two levels—zero and one transfer. To avoid directing travellers attention to this aspect of transfers, we did not explicitly explain the potential for increased exposure in the survey.

*Policy-based attributes*. 3. Sanitization: At the beginning of the pandemic it was suspected that the virus could also spread via surfaces. In response to advisories based on this information, some public transport operators increased their sanitization regime; for example, the New York City subway was disinfected nightly [37]. In what some critics have called 'hygiene theatre' [38], such disinfection efforts continued even after it became known that airborne respiratory droplets were the primary mode of transmission [39] possibly to maintain a sense of security. In line with the government's initial 'soft' approach to the virus, operators in the Netherlands did not change their cleaning protocols drastically. We test here if perceived risk would reduce if travellers know that their trains are cleaned more frequently. Thus, two levels are included: the current status quo (no extra sanitization) and extra cleaning rounds during the day. The current regime is noted in the survey to indicate what extra sanitization involves.

4. Face mask policy: Face masks have been found to be an effective method for preventing spread of the disease [40] and mandatory face masks could, as such, mitigate perceived risk. Since social distancing is not possible aboard public transport vehicles, face masks have been made mandatory in many public transport systems, including those in the Netherlands. However, face masks remain a contentious issue in the Netherlands [41] and even those concerned about the virus seem to be particularly reluctant to use them [30]. We test here if face mask mandates are successful in reducing perceived risk among Dutch train travellers. This attribute is varied as a binary variable: face masks are mandatory or not for all people above 13. We specified that masks were only required inside vehicles in line with regulations imposed at the time of executing the experiment.

5. Lockdown status: Since early 2020, governments around the world have imposed varying degrees of lockdowns to limit COVID-19 spread. While some may feel more secure with firm government intervention [42], the level of restrictions set may further impress the seriousness of the situation upon others. For the experiment, we used four levels: 'normal life' with no

restrictions whatsoever (pre-pandemic); 'social-distancing' where only a few advisories, such as social distancing and increased hand washing, are issued; the 'moderate lockdown' (imposed October 2020) in which large-scale events are forbidden and capacity limits have been set; and the so-called (by the Dutch government and consequently in the news media) 'intelligent lockdown' (imposed March 2020) where, in addition to the previous level, working from home is strongly advised, dining and entertainment venues are closed, and a maximum of three home visitors are allowed.

*Pandemic-context attributes.* 6. Infection rate: Unlike other attributes, infection is an exogenous risk factor which cannot be directly influenced. As noted in our literature review, reported cases has been found to be an important predictor for public transport use during past pandemics. Similar to Shelat et al. [30], we define infection rate as the number of contagious individuals in the country for which estimates are available from the national health body [43]. Although the number of contagious people is always an estimate, this metric is selected because it is independent from changes in testing capacity and willingness. To assist interpretation of this variable, the time period where the Netherlands had a similar infection rate is also presented. In the experiment, we include four levels: 20 per 100,000 (July 2020); 600 per 100,000 (October 2020—2nd peak); 1000 per 100,000 (March 2020—1st peak); and 10,000 per 100,000 (not observed, extremely high).

An orthogonal design is used to create 12 attribute profiles which are blocked into two parts to limit respondent burden. Respondents are randomly assigned to either block. Fig 2 shows the example question shown to respondents after an explanation of each of the attributes included.

**3.2.2. Route choice.** The second part of the experiment is the choice-based bridging experiment where respondents first choose between two train alternatives and then state whether they would rather opt-out (Fig 3). As discussed previously, the train alternatives are characterised by travel cost, travel time, and perceived risk. To increase familiarity with the choice situations, separate experiments are designed for respondents usually making short (less than 30 minutes) and long trips, respectively. Moreover, respondents were informed that they were travelling for the 'usual' purpose for which they used the trains before the pandemic.

Three travel time levels are included: 10, 17, 24 minutes for the short-trip experiment and 35, 45, 55 minutes for the long-trip experiment. A wider range is included for the longer trip to account for potential concavity of the utility function, that is, 10 additional minutes on an already long trip may be perceived to be less taxing than 10 minutes on a shorter trip. After an analysis on the relationship between travel times and ticket prices of train trips in the Netherlands, we set the travel cost attributes as €3, €4.5, €6 and €9, €12, €15 for short- and long-trip experiments, respectively. We ensured that all time-cost combinations existed for actual trips. Respondents were asked to disregard any discount cards they may have. The perceived risk attribute was presented to respondents as 'your infection risk rating' and they were informed that this was the risk rating they would have given to the train alternative. The attribute was presented on the same 5-point scale as the rating experiment and for both versions of the choice experiment, three levels were included: very low, medium, and very high risk.

Given the pandemic context of this choice experiment, it is not unlikely that some respondents may not wish to travel by trains at all [30]. Therefore, after a choice between two train alternatives have been made, respondents can indicate that they would rather not choose either option. We force the former choice to enable estimation of trade-offs in case the majority chooses to opt-out. At the end of the choice experiment, respondents were asked which of the following they would do if they opted out in any of the foregone scenarios: using another mode (car, bicycle, or other), working/studying from home, or cancelling the activity.

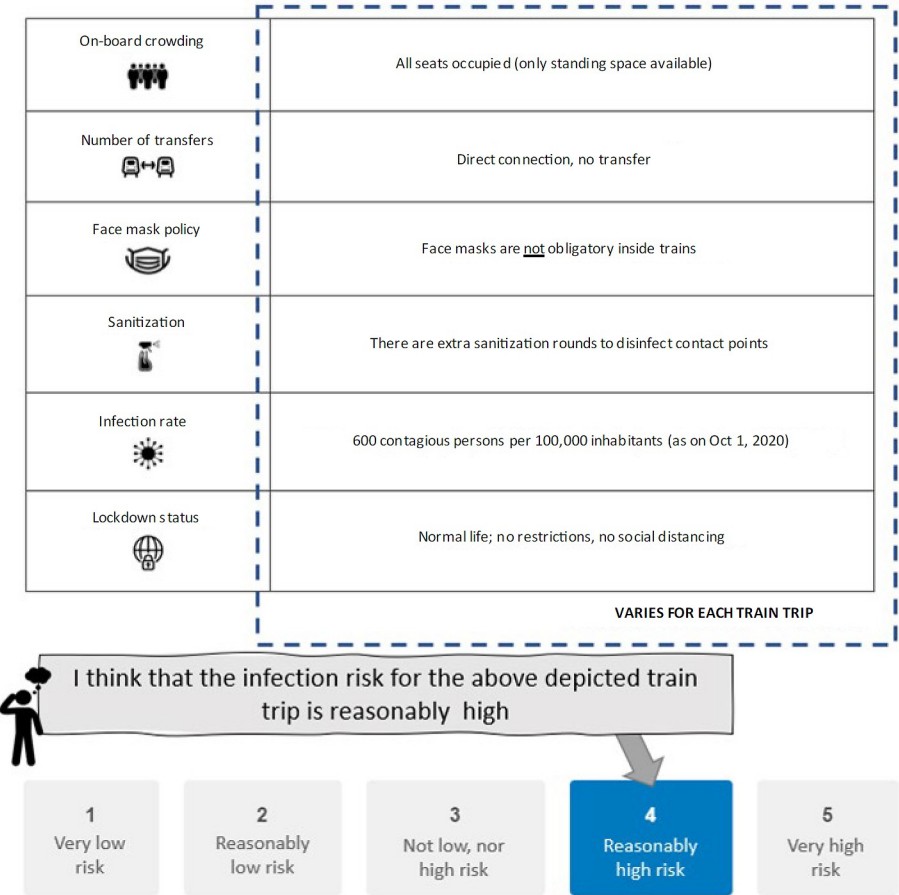

**Fig 2. Screenshot of the example task in the rating-based sub-experiment (translated to English).**

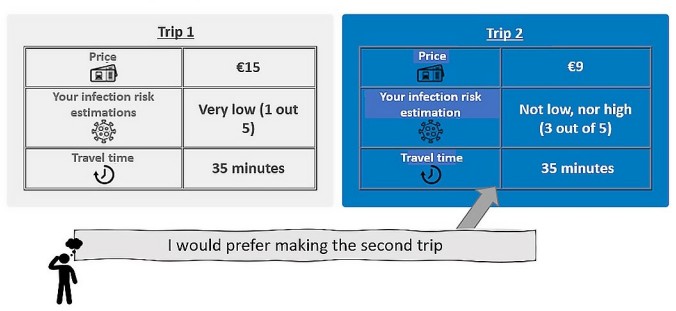

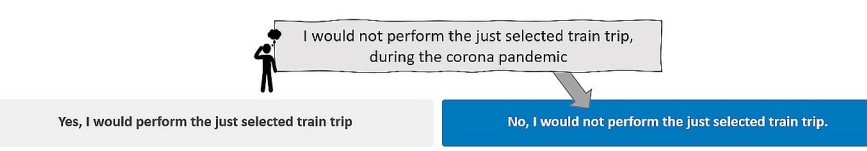

**Fig 3. Screenshot of the example task in the choice-based bridging experiment (translated to English).**

For the final experiment, a D-efficient design is applied. Priors for this design are obtained from an MNL model estimated for a pilot survey with 56 respondents. The pilot also allowed us to ensure that the full range of risk ratings were observed, perform a sanity check on parameter values, and receive general feedback. For each version of the experiment (short-/long-trip), nine choice scenarios were created. Both the choice-based bridging experiment and the rating-based sub-experiment were generated with Ngene [44].

### 3.3. Data

In addition to the risk perception and route choice experiments, three categories of personal characteristics were also collected: (i) mobility-related factors, (ii) socio-demographic factors, and (iii) COVID-19 related qualitative measures. In the first category, we asked travellers how often they used the train before and during the pandemic-related restrictions, their most frequent purpose of travel when using trains, car availability, and typical train trip duration. These mobility characteristics were only used for the bridging choice experiment. In the second category, we asked for their age, gender, education, and employment status. The predictors in the final category are mostly Likert scale measures and were selected from [45] and [46]. The full list of COVID-19 related qualitative indicators can be found under S1 Table.

The final survey was distributed using an online panel to Dutch train travellers who made at least six trips per year before the pandemic. Data collection took place in the first week of December 2020, just before the imposition of a stricter lockdown. The survey was offered in Dutch and had an expected completion time of 12–15 minutes. A data management plan for the survey and the related informed consent was approved by the Human Research Ethics Committee of Delft University of Technology. Respondents were informed that they participated voluntarily, could stop the survey at any time, and that provided information would be used for research and subsequent scientific publication. They were assured that answers were anonymous. A total of 513 responses were collected of which 408 are considered to be valid. Responses were eliminated from further analysis if they were incomplete or had a fill-in time shorter than 5 minutes. To ensure that the sample was representative of the Dutch train user population, we set quotas for age and gender. Desired stratifications were obtained from the data collected in the national, one-day, trip diary survey, OViN (*Onderzoek Verplaatsingen in Nederland*) between 2011 and 2015 [47]. Table 1 shows a selection of sample characteristics. Note that choice experiment observations are skewed towards commuting travel purposes and long trips.

### 3.4. Analysis

**3.4.1. Risk perception.** The risk perception decision construct is analysed using ordinary linear regression. Strictly speaking, since the risk rating is measured on an ordinal scale, an ordinal model is implied. However, as noted in [33], this would mean that risk perception should also be treated as an ordinal variable in the bridging experiment and entered as a set of dummy variables. Since this would increase the number of variables to be estimated, we choose to consider the rating as an interval level measurement. As shown in Fig 1, risk perception is dependent on the previously discussed risk attributes and personal characteristics (socio-demographic and COVID-19 related). In addition to main effects, a selection of quadratic terms and relevant interaction effects are also tested. The regression is performed in SPSS using step-wise backwards elimination (parameters with a p-value > 0.05 are considered insignificant), prioritizing main effects over interaction variables.

**3.4.2. Route choice.** We adopt the random utility maximization framework where the utility of alternative *i* for individual *n*, $U_{in}$, consists of a systematic component ($V_{in}$) and a

**Table 1. Sample characteristics.**

| Total respondents | 408 | | |
|---|---|---|---|
| | | Distribution (%) | |
| Attribute | Value | Actual | Required |
| Gender | Female | 46% | ~50% |
| | Male | 54% | ~50% |
| | Other | 0% | |
| Age | 18–24 | 28% | 36% |
| | 25–34 | 18% | 17% |
| | 35–44 | 14% | 13% |
| | 45–54 | 17% | 16% |
| | 55–64 | 15% | 12% |
| | >64 | 7% | 6% |
| Trip purpose | Work | 42% | |
| (before pandemic) | Education | 18% | |
| | Leisure | 40% | |
| Trip length | < 30 minutes | 29% | |
| (before pandemic) | > 30 minutes | 71% | |

random ($\varepsilon_{in}$) component. The systematic component is assumed to be linear additive, computed as the sum of the alternate specific constant ($\beta_i$) and the product of taste preferences ($\beta_{ij}$) and the values of attributes, $j$ ($x_{ijn}$) (Equation [1]). Both multinomial logit and mixed multinomial logit models were estimated but we ultimately chose the latter as they accounted for the panel structure in the data and substantially improved model fit. In random parameter mixed logit, when the parameter for attribute $j$ is considered random, the likelihood of observing sequence of choices, $i^*$, in choice situations, $t$, by an individual is calculated as shown in Equation [2] (for each random parameter, an additional integral is required). Since this does not have a closed form solution, it is estimated using maximum simulated log-likelihood with quasi-random Halton draws. The number of draws is determined by doubling the number of draws until convergence (in terms of consistency of the parameter values and model fit) is reached. Separate models are estimated for the short- and long-trip experiments using a variety of utility specifications (e.g., testing for non-linear and interaction effects, different combinations of fixed and random taste parameters, and different distributions for the latter). The estimation procedure is similar to the rating experiment: insignificant (p-value < 0.05) parameters are removed sequentially until only significant parameters remain in the final model. All models were estimated using Apollo [48].

$$U_{in} = V_{in} + \varepsilon_{in}$$
$$V_{in} = \beta_i + \sum_j \beta_{ij} \cdot x_{ijn} \qquad [1]$$

$$P_{i*nt}(\beta_j) = \frac{e^{V_{in}(\beta_j)}}{\sum_{i'}^{I} e^{V_{i'n}(\beta_j)}}$$
$$L_n = \int_{\beta_j} \left[ \prod_t P_{i*nt}(\beta_j) \right] f(\beta_j) \mathrm{d}\beta \qquad [2]$$

## 4. Results

In this section, we first present results of the sub- and bridging experiment, and then combine these models to produce willingness-to-pay metrics for various risk mitigating factors and develop scenarios for travel demand. Table 2 shows an overview of all the attributes included in the final models.

### 4.1. Risk perception

Table 3 shows the final ordinary linear regression model for perceived risk. Since interaction effects only improves the model fit incrementally (adjusted $R^2$ improves from 0.341 to 0.348), we present the model with only main effects. Higher on-board crowding and infection rates increase the perception of risk while mask mandates and increased sanitization put travellers at ease. These findings are in line with those reported by Aaditya and Rahul [32] and Shelat et al. [30, 32]. The former pair of attributes also have the largest effect on risk perception. Non-linear effects in these attributes are not prominent and do not improve model fit. We also do

**Table 2. Overview of attributes included in the final models.**

| Attributes | Measurement Level[a] | Description and Levels |
|---|---|---|
| **Risk attributes** | | |
| On-board crowding | Ratio | 1 (10% occupancy), 3 (30% occupancy), 6 (60% occupancy), 10 (100% occupancy) |
| Transfers | Categorical | Zero (ref.), One |
| Sanitization | Categorical | No extra sanitization (ref.), Extra sanitization rounds |
| Face mask policy | Categorical | No mandate (ref.), Masks compulsory on-board |
| Lockdown status | Categorical | No restrictions (ref.), Social distancing, Moderate lockdown, Intelligent lockdown |
| Infection rate | Ratio | 0.2 (20 per 100,000), 6 (600 per 100,000); 10 (1,000 per 100,000); 100 (10,000 per 100,000) |
| **Train Attributes** | | |
| Travel cost | Ratio | Short-trip (3, 4.5, 6); Long-trip (9, 12, 15) in € |
| Travel time | Ratio | Short-trip (10, 17, 24); Long-trip (35, 45, 55) in minutes |
| COVID-19 risk perception | Ordinal | 1 (very low)–5 (very high) |
| **Personal characteristics** | | |
| Personal efficacy | Ordinal | The actions that I personally take to prevent the spread of the coronavirus (e.g., by limiting the number of social contacts, washing hands, wearing a face mask, etc.) are effective. |
| | | Completely disagree (1)–Completely agree (5) |
| Perceived control | Ordinal | Overall, I believe that I can control or avoid becoming infected by the coronavirus (e.g., by limiting social contact, washing hands, wearing a face mask, etc.). |
| | | Completely disagree (1)–Completely agree (5) |
| Personal health anxiety | Ordinal | I never worry about my health (1)–I almost always worry about my health (4) |
| Vulnerability of close ones | Ordinal | Overall, I believe that people that I care about (e.g., grandparents) are at risk of becoming infected and seriously ill due to the coronavirus outbreak. |
| | | Completely disagree (1)–Completely agree (5) |
| COVID-19 experience | Categorical | Do you know someone who attained the coronavirus? Now or in the past? |
| | | No (ref.), Yes |
| Age | Ratio | 18–87 |
| Education level | Categorical | Lower (vocational diploma or lower) (ref.), Higher |
| Employment status | Categorical | Other/rather not say (ref.), Employed, Unemployed, Student, Retired |
| Train use (during pandemic) | Categorical | Lower (< 1 day/week) (ref.), Higher (≥ 1 day/week) |

[a]All ordinal variables are treated as interval level measurements. Categorical variables are all effect coded ('ref.' indicates the reference level).

**Table 3. Risk perception model.**

| | Model | Ordinary Linear Regression | | |
|---|---|---|---|---|
| | # Parameters | 11 | | |
| | Adjusted $R^2$ | 0.337 | | |
| | | Coeff. | Stand. coeff. | p-val |
| | Constant | 1.566 | | < 0.001 |
| **Risk attributes** | On-board crowding | 0.117 | 0.346 | 0.001 |
| | Number of transfers | - | - | - |
| | Sanitization | -0.144 | -0.126 | < 0.001 |
| | Face mask policy | -0.216 | -0.189 | < 0.001 |
| | Moderate lockdown | -0.185 | -0.114 | < 0.001 |
| | Infection rate | 0.007 | 0.243 | < 0.001 |
| **COVID-19 related characteristics** | Perceived efficacy | 0.174 | 0.128 | < 0.001 |
| | Perceived control | -0.122 | -0.094 | < 0.001 |
| | Personal health anxiety | 0.133 | 0.074 | < 0.001 |
| | Vulnerability of close ones | 0.168 | 0.134 | < 0.001 |
| **Socio-demographic** | Student | -0.087 | -0.036 | 0.030 |

not find a significant effect of the number of transfers. The fact that travellers' risk perception is lowered with a moderate lockdown (as compared to no restrictions) indicates that some level of governmental intervention can indeed have a positive effect on risk perception as Oude Groeniger et al. [42] suggest. We further hypothesised that 'doing your own research' by voluntarily accessing information would lead to more critical viewpoints regarding preventive measures but significant interactions effects between media consumption and mask or sanitization policies are not found.

Amongst COVID-19 related measures, we find that low perceived control and high vulnerability increases perceived risk. The fact that higher levels of perceived control usually go hand in hand with lower perceived risks has also been established outside the context of the pandemic [49]. In light of the inverse relationship between perceived control and risk perception, we note that perceived efficacy has an unexpected sign and offer the following explanation. There may be a direct association between people who appreciate the risks posed by the virus and those who believe in the effectiveness of recommended measures to stop the spread. However, perceived efficacy (at the societal level) will not perfectly translate to perceived control (at the individual level) for all of these people, particularly risk averse individuals. Finally, similar to Mertens et al. [46], we also find that vulnerability of close ones is a very strong predictor of the level of perceived COVID-19 risk. In fact, we find that it is about twice as important as personal health. About 69% of the respondents report knowing someone who has had COVID-19. On average, this experience increases perceived risk.

We do not find any significant impacts of socio-demographics except that students have a lower risk perception *ceteris paribus*. Given that vulnerability objectively increases with age, one would expect it to impact risk perception. However, previous studies have also found age to not be a reliable indicator for COVID-19 risk perception [17, 45, 46]; potentially because of other interacting variables, such as information sources. The increased risk with age may have also been represented more directly by the qualitative measure of personal health anxiety. Lastly, in contrast to our results, a number of studies in various domains (including COVID-19) have found that female respondents perceived risks to be greater [50].

## 4.2. Route choice

For each scenario in the route choice experiment, respondents choose between two train alternatives and then indicate whether they would rather opt-out. At the end of the experiment, respondents also choose their preferred opt-out option. The train alternatives are characterised by travel cost, travel time, and COVID-19 risk perception. While we do not include exposure duration as a risk attribute for reasons mentioned previously, we test for the effect by including an interaction effect between travel time and risk perception. For the opt-out alternative, apart from an alternative-specific constant (representing the penalty of opting out), we also test the effect of different opt-out scenarios by including them as categorical variables. Finally, the effects of various mobility and socio-demographic factors either directly on the opt-out utility or as interaction effects with the train alternative attributes are estimated.

In the mixed logit models, we tried estimating random parameters for travel cost, travel time, and risk perception, using normal, lognormal, and triangular distributions. Since these parameters are expected to be negative, normal distributions, being unbounded on both sides, will result in counter-intuitive values to some extent. However, as the bounded distributions resulted in unexpected parameters, we use normal distributions nevertheless. Random parameters were only meaningful for travel cost and risk perception (estimated standard deviations for travel time were insignificant). In the final models, less than 0.3% and 12% of the probability masses of travel cost and risk perception parameters, respectively, have the wrong sign.

Table 4 shows the final mixed logit models for the short- and long-trip route choice experiments. All main parameters including the opt-out constant have the expected signs, indicating that travellers are likely to choose routes with lower travel time, cost, and perceived infection risk. Both travel purpose and opt-out scenario did not have a significant effect in either model. For the long-trip model, we find that younger travellers and those reporting higher train use during the pandemic put a lower weight on risk perception. This is in line with the results in

**Table 4. Route choice model.**

| Model | Short-trip Mixed Logit | | Long-trip Mixed Logit | |
|---|---|---|---|---|
| # Respondents | | 117 | | 291 |
| # Parameters | | 10 | | 11 |
| Initial LL | | -1148.05 | | -2869.58 |
| Final LL | | -997.11 | | -1777.75 |
| Adjusted $\rho^2$ | | 0.127 | | 0.377 |
| | **Coeff.** | **p-val** | **Coeff.** | **p-val** |
| Travel cost mean | -0.501 | < 0.001 | -0.399 | < 0.001 |
| Travel time | -0.104 | < 0.001 | -0.054 | < 0.001 |
| Risk perception mean | -0.616 | 0.005 | -1.258 | < 0.001 |
| Opt-out constant | -7.703 | < 0.001 | -11.538 | < 0.001 |
| Travel cost std. dev. | 0.163 | < 0.001 | 0.067 | < 0.001 |
| Risk perception std. dev. | 0.507 | < 0.001 | 1.123 | < 0.001 |
| Student × Travel cost | | | 0.080 | 0.018 |
| Student × Travel time | | | -0.029 | 0.002 |
| Age × Risk perception | | | -0.015 | 0.005 |
| Highly educated × Risk perception | 0.158 | 0.020 | -0.224 | 0.013 |
| Employed × Risk perception | -0.457 | 0.049 | | |
| Student × Risk perception | -0.646 | 0.012 | | |
| Retired × Risk perception | -0.768 | 0.002 | | |
| High train use (during pandemic) × Risk perception | | | 0.273 | 0.005 |

Shelat et al. [30] who also report over-representation of these groups in their 'Infection Indifferent' latent class. In the same model, students are found to care less about time and more about cost although we would normally expect the opposite effect. The latter may be explained if we consider that (despite explicit instructions to the contrary), student respondents took into account their free public transport cards when making choices. In the short-trip model, students place more importance on risk perception than employed individuals but less than retired travellers. Note that in the ratings experiment, being a student is associated with lower risk perception. Finally, level of education is, surprisingly, found to have opposite effects on the importance of risk perception in the two models.

Since our models have random parameters for travel cost and risk perception, we use simulation to calculate values of travel time and risk perception. For each random parameter, we make 100,000 draws from its distribution, add the relevant interaction effects to each draw, and then calculate the required ratios for each (pair of) draw(s). From the distributions thus obtained, we report the median values as they are not affected by extremely large or small draws. In the interest of conciseness, for the route choice model and the following integrated model, we present median willingness to pay values only for an 'average' respondent in our sample. To do this we ignore all interaction variables with effect-coded attributes and use the median value of numerical ones. Thus, we only need to account for age (median = 37) in the long-trip model.

For the short- and long-trip models, we find values of time equal to €12.46 and €8.11 per hour, respectively. We find assurance in the fact that our values of time are close to the €11/ hour (converted from €9.25/hour in 2010 to 2020 values considering inflation) value found in an extensive study in the Netherlands [51]. The higher value of time in the short-trips group, potentially confirms our expectations regarding a concave utility for travel time. Although we did not find a significant parameter for the interaction of travel time and risk perception, we do find a higher value of perceived risk for the longer-trip group in general. We estimate that the average respondent making a long trip is willing to pay €4.54 to reduce the risk rating by one point whereas for a short trip they are willing to pay only €1.24. A possible reason for this difference in willingness to pay may be that although travellers' risk trade-offs are not affected by the range of travel times offered within each experiment, there is some recognition that longer trips increase exposure. Without further testing, we cannot exclude the possibility that those who make longer trips simply tend to be more risk averse.

### 4.3. Willingness to pay

The ability to combine results from the bridging experiment and sub-experiments is integral to the hierarchical information integration approach. Using the risk attribute coefficients from the linear regression model and the (median) value of perceived risk from the choice model, we can calculate the willingness to pay for various risk-reducing factors. This is given by the product of the risk attribute's marginal contribution to risk perception and the value of perceived risk. Such willingness to pay values could then be used, for example, in cost-benefit analyses. The value of crowding is an important parameter for public transport planning, particularly during a pandemic. We find that respondents in the short- and long-trip groups are willing to pay €0.14 and €0.53 for a reduction of seat occupancy by 10 percentage points. Obtaining comparable values from literature is challenging for two reasons. First, previous studies have expressed willingness to pay for crowding through a variety of definitions and units, ranging from time multipliers to per minute values to values based on probability of having to stand [36]. Second, since most estimates are pre-pandemic, they have focused on higher load factors (typically >80%) for which travellers were expected to perceive an additional cost.

In contrast, due to the COVID-19 pandemic, travellers are likely to be wary of crowding already at a much lower occupancy. Instead, we put the values of crowding in context of the available options. Upgrading to the first class compartment in the Dutch railways costs an extra €8 for a trip of about 45 minutes. Amongst other amenities (e.g., wider seats), first class compartments are typically less crowded. By comparison, our model for trips longer than 30 minutes indicates that an average respondent is willing to pay about €1.60 to reduce occupancy from 60% (unable to sit alone) to 30% (able to sit alone). Other mitigation factors include increased sanitization and face mask enforcement. For short- and long-trips, respectively, travellers are willing to pay €0.18 and €0.65 extra per trip for increased sanitization and €0.27 and €0.98 extra per trip for on-board mask mandates.

## 5. Discussion and conclusions

Unprecedented changes to travel behaviour have followed the outbreak of the COVID-19 pandemic since early 2020. Public transport use, in particular, has suffered significantly. Authorities' desire to contain the virus with public transport restrictions early on in the crisis and intermittent lockdowns dictated by case numbers have directly affected the ability and willingness to travel. In addition to conventional travel attributes, such as time and cost, travellers are likely to now contemplate another variable: infection risk. Perceptions regarding infection risk may have been critical to the decline in mode share that public transport networks have suffered and are likely to continue evolving along with the pandemic crisis and during its aftermath. Therefore, to ensure that the shift to private modes is not a lasting effect of the pandemic, authorities and operators need to better understand these perceptions as well as mitigating factors. Although a large body of literature exists on COVID-19 infection risk perceptions and changes in travel patterns since the outbreak, few have explicitly analysed risk perception in public transport and its impact on behaviour within the system.

In this study, using a two-stage stated choice experiment, we first quantified the impact of different public transport and contextual factors on travellers' COVID-19 risk perception, and then evaluated the trade-off between this risk perception and other travel attributes. The stated choice experiment was distributed to train travellers in the Netherlands in December 2020; just before the imposition of the strictest lockdown (in response to rapidly rising case numbers) and before vaccines against the virus were available. Applying a common variant of the conventional hierarchical information integration approach, we first asked respondents to rate how risky they perceive various public transport options to be based on six trip-, policy-, and pandemic-related attributes. Then, in a bridging experiment, we observed choices between different route alternatives that include their own risk rating as an attribute in addition to travel time and cost. Depending on their typical train trip length, respondents faced one of two bridging experiment versions.

Given the rapidly changing nature of the pandemic, travellers' risk perception is likely to be affected by not only direct features such as infection rates but also indirect ones such as lockdowns and mask mandates. Therefore, it is critical that we account for a variety of contextual factors especially while the pandemic looms large on travellers' minds. The hierarchical information integration approach, not only aligns conceptually with such decision problems but also allows us to evaluate the impact of a larger number of variables without overloading respondents. Thus, although we only capture a snapshot of the behavioural response to this fast-changing situation, we can use the estimated choice model to predict behaviour under detailed pandemic scenarios and assess the benefits of various risk mitigating policies. Furthermore, we noted several similarities between the results from this two-stage experiment and earlier single-stage studies, in particular [30].

We modelled the perceived risk as a linear regression of risk attributes from the ratings experiment, responses to qualitative COVID-19 factors, and socio-demographic characteristics. On-board crowding and infection rates are the most important factors for risk perception. Despite the national debate over face masks and an apparent dislike for them [41], mask mandates are perceived as the most important mitigating factor. We also noted that vulnerability of family and friends had more than twice the impact on increasing perceived risk than personal health risks. Next, from the choice observations in each bridging experiment version, we estimated mixed logit models with normally distributed random parameters for travel cost and perceived risk. As anticipated, the choice models revealed that respondents preferred routes not only with lower travel times and cost but also with a lower perceived COVID-19 risk. The estimated values of time are in line with a pre-pandemic appraisal in the Netherlands. Respondents in the shorter trip experiment value time fifty percent more than those in the longer trip experiment, confirming a concave utility curve for travel time. In contrast, long trip travellers are willing to pay nearly four times as much as their short trip counterparts for an option they perceive to be less risky. While interactions between travel time and risk were not significant within either experiment, the difference in value of risk across the two groups might indicate that there is indeed some recognition that longer trips increase exposure. By combining the risk perception and route choice model, we estimated willingness to pay for different risk mitigating factors. The amount travellers are willing to pay for increased sanitization, mask enforcement, and crowd reduction may be used for a cost-benefit analysis for mitigating strategies. We do note that the current alternative available for crowding avoidance (i.e., upgrading to first class) is found to be much more expensive than the (average) willingness to pay.

In a sign of how quickly the situation is evolving, a number of new risk-determining factors may have become important since we conducted our survey. The most prominent of these are related to COVID-19 vaccinations and new variants of the virus. Health authorities around the globe have cleared vaccinations for emergency use since late 2020. With this important development, one's vaccination status, the proportion of vaccinated population, and vaccine mandates and checks are likely to become critical risk mitigating factors. On the other hand, perceived risk can increase when variants of concern are borne out of virus mutations. New variants may spread more easily, be more deadly, and even threaten to break through the protections offered by vaccines. The effect of a new variant, the geographical region where it is first detected, and consequent local countermeasures and media reporting could all affect perception.

Despite the scientific attention it has drawn, peer-reviewed studies often lag behind rapidly developing phenomena such as the COVID-19 pandemic. We would like to stress that such studies are, nevertheless, important. First because, given the emergence of coronavirus variants and large swathes of populations without access to vaccines, we are not out of the current pandemic. Second and more importantly, we can be sure that this pandemic will not be the last of its kind. Therefore, although this study may be missing (what currently seem to be) critical factors, they provide important results regarding traveller preferences in key stages of the pandemic. Naturally, with the widespread availability of vaccines in many parts of the world and subsequent changes in local regulations, it is also important that we keep updating our understanding of travellers' risk perceptions and behaviour.

## Supporting information

**S1 Table. Questions used for COVID-19 related qualitative indicators (in English).**
(DOCX)

## Author Contributions

**Conceptualization:** Sanmay Shelat, Thijs van de Wiel.

**Formal analysis:** Thijs van de Wiel.

**Funding acquisition:** Oded Cats.

**Investigation:** Thijs van de Wiel.

**Methodology:** Thijs van de Wiel, Eric Molin.

**Supervision:** Sanmay Shelat, Eric Molin, J. W. C. van Lint, Oded Cats.

**Writing – original draft:** Sanmay Shelat.

**Writing – review & editing:** Eric Molin, J. W. C. van Lint, Oded Cats.

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
