## [Decision Letter · Decision Letter 0]

6 Dec 2021

PONE-D-21-35962Analysing the impact of COVID-19 risk perceptions on route choice behaviour in train networksPLOS ONE

Dear Dr. Shelat,

Thank you for submitting your manuscript to PLOS ONE. After careful consideration, we feel that it has merit but does not fully meet PLOS ONE’s publication criteria as it currently stands. Therefore, we invite you to submit a revised version of the manuscript that addresses the points raised during the review process.

We look forward to receiving your revised manuscript.

Kind regards,

Jian Wang

Academic Editor

PLOS ONE

Journal Requirements:

" ext-link-type="uri" xlink:type="simple">https://journals.plos.org/plosone/s/file?id=ba62/PLOSOne_formatting_sample_title_authors_affiliations.pdf"

Reviewers' comments:

Reviewer's Responses to Questions

**Comments to the Author**

1. Is the manuscript technically sound, and do the data support the conclusions?

Reviewer #1: Yes

Reviewer #2: Partly

2. Has the statistical analysis been performed appropriately and rigorously? 

Reviewer #1: Yes

Reviewer #2: Yes

3. Have the authors made all data underlying the findings in their manuscript fully available?

Reviewer #1: Yes

Reviewer #2: Yes

4. Is the manuscript presented in an intelligible fashion and written in standard English?

Reviewer #1: Yes

Reviewer #2: Yes

5. Review Comments to the Author

Reviewer #1: This manuscript presents a two-stage stated choice experiment. The first stage quantifies the impact of contextual factors on travellers’ COVID-19 risk perception, and the second stage analyzes the route choice behaviour by evaluating the trade-off between this risk perception and common travel attributes. This manuscript has a clear structure and takes research on the impact of COVID-19 on travel behaviours, especially for those related to public transportation. Some aspects should be improved and a few detailed comments are listed as follows:

1. I found the methods used in this manuscript are ordinary, that is, linear regression for analyzing the risk perception and multinomial logit models for evaluating the route choice. Researchers have proposed multiple models on correlation analysis and the discrete selection, and used them to make research related to COVID-19. However, this manuscript ignores the summary of existing models and fails to explain why these models are chosen or what irreplaceable advantages do they have over other models.

2. In Section 3.2.1, some classification levels for observed attributes of COVID-19 risk perception seem unreasonable in the rating-based sub-experiment. Firstly, the name “intelligent lockdown” is misleading and it should be further distinguished from the “moderate lockdown”; Secondly, levels of infection rate are too close between 6 per 1000 and 10 per 1000, even if they were observed from actual peaks. It may be hard for respondents to distinguish these two levels and I prefer to see them as the same level, compared to 0.2 per 1000 and 100 per 1000.

3. In Table 2, I can’t understand why the type of travel cost, travel time, and age is a ratio. The authors need to check if there are mistakes or not. If not, this manuscript is encouraged to give more descriptions in this table or the text.

4. This manuscript mentioned that the two-stage experiment conceptualized COVID-19 related variables as affecting travellers’ perceived risk, in turn, affects their behaviour. I agree with this idea, however, some attributes such as infection rate can represent the level of risk perception. Thus, I expected to see a discussion on the discrete selection model with these attributes as direct input and further evaluate its difference from this two-stage model. Maybe some interesting conclusions can be drawn and the advantages/disadvantages of this two-stage model can be better represented.

5. There are some minor problems that should be corrected:

1) Format error at the beginning of Section 4.1, see “Error! Reference source not found.”

2) All figures in the manuscript is blurring and should be replaced with high-quality figures;

3) The sum of distribution of actual age levels in Table 1 should reach 100%;

Reviewer #2: The title of this paper is “Analysing the impact of COVID-19 risk perceptions on route choice behaviour in train networks”. To my understanding, this paper should focus on studying the impact of COVID-19 risk perceptions on route choice behaviour. However, a large part of this paper is explaining risk perception and the impact of economic policies and so on. The impact of COVID-19 risk perceptions on route choice behaviour is not clear. Authors should add more explanations on route choice research and provide more related results.

6. PLOS authors have the option to publish the peer review history of their article (what does this mean?). If published, this will include your full peer review and any attached files.

Reviewer #1: No

Reviewer #2: No

---

## [Author Response · Author response to Decision Letter 0]

17 Jan 2022

Dear Dr Jian Wang and the Editorial Board of PLoS One,

We would like to thank you and the two reviewers for their constructive comments and suggestions. In the attachment, we present responses to individual reviewers’ comments (which we have numbered for reference) and outline all the changes made to meet journal requirements.

Should the editor or the reviewers have any further questions, we will gladly provide further clarification.

Sincerely yours,

Sanmay Shelat, Thijs van de Wiel, Eric Molin, Hans van Lint, Oded Cats

---

## [Decision Letter · Decision Letter 1]

17 Feb 2022

Analysing the impact of COVID-19 risk perceptions on route choice behaviour in train networks

PONE-D-21-35962R1

Dear Dr. Shelat,

We’re pleased to inform you that your manuscript has been judged scientifically suitable for publication and will be formally accepted for publication once it meets all outstanding technical requirements.

Kind regards,

Jian Wang

Academic Editor

PLOS ONE

Additional Editor Comments (optional):

Reviewers' comments:

Reviewer's Responses to Questions

**Comments to the Author**

1. If the authors have adequately addressed your comments raised in a previous round of review and you feel that this manuscript is now acceptable for publication, you may indicate that here to bypass the “Comments to the Author” section, enter your conflict of interest statement in the “Confidential to Editor” section, and submit your "Accept" recommendation.

Reviewer #1: All comments have been addressed

Reviewer #2: All comments have been addressed

2. Is the manuscript technically sound, and do the data support the conclusions?

Reviewer #1: Yes

Reviewer #2: Yes

3. Has the statistical analysis been performed appropriately and rigorously? 

Reviewer #1: Yes

Reviewer #2: Yes

4. Have the authors made all data underlying the findings in their manuscript fully available?

Reviewer #1: Yes

Reviewer #2: Yes

5. Is the manuscript presented in an intelligible fashion and written in standard English?

Reviewer #1: Yes

Reviewer #2: Yes

6. Review Comments to the Author

Reviewer #1: The comments have been addressed. The revised manuscript corrected some mistakes and added some explanations.

No additional comments.

Reviewer #2: Auntors have revised the paper according to my advice. I am glad to recommend this paper to be accepted.

7. PLOS authors have the option to publish the peer review history of their article (what does this mean?). If published, this will include your full peer review and any attached files.

Reviewer #1: No

Reviewer #2: No